# Bio-Model Selection, Processing and Results for Bio-Inspired Truck Streamlining

**DOI:** 10.3390/biomimetics8020175

**Published:** 2023-04-23

**Authors:** Xiaoyin Fang, Eize J. Stamhuis

**Affiliations:** Biomimetic Group, Energy and Sustainability Research Institute Groningen, Faculty of Science and Engineering, University of Groningen, 9747 AG Groningen, The Netherlands

**Keywords:** bio-inspired design, streamlined shape, CFD, drag resistance

## Abstract

We introduce a method for the selection and processing of a biological model to derive an outline that provides morphometric information for a novel aerodynamic truck design. Because of the dynamic similarities, our new truck design will be inspired by biological shapes with a known high level of streamlining and low drag for operation near the seabed, i.e., the head of a trout, but other model organisms will also be used later. Demersal fish are chosen because they live near the bottom of rivers or the sea. Complementary to many biomimetic studies so far, we plan to concentrate on reshaping the outline of the fish’s head and extend it to a 3D design for the tractor that, at the same time, fits within EU regulations and maintains the truck’s normal use and stability. We intend to explore this biological model selection and formulization involving the following elements: (i) the reason for selecting fish as a biological model for a streamlined truck design; (ii) The choice of a fish model via a functional similarity method; (iii) biological shape formulization based on the morphometric information of models in (ii) outline pick-up, a reshaping step and a subsequent design process; (iv) modify the biomimetic designs and test utilizing CFD; (v) further discussion, outputs and results from the bio-inspired design process.

## 1. Introduction

Goods in Europe are mostly transported by heavy trucks, and a large proportion of these are articulated, having a tractor and a trailer. Fuel consumption makes up a large amount of the costs related to daily transportation, and the resulting air pollution creates many environmental risks, e.g., human respiratory diseases and high CO_2_ emissions, which ultimately increase the greenhouse effect [1,2,3,4]. Urban delivery trucks with a 4 × 2 axle configuration (4-UD) emit, on average, 307 g CO_2_/t-km, which is over five times as much as long-haul tractor–trailers (5-LH), with emissions of 57 g CO_2_/t-km [5]. Therefore, the unit used, g CO_2_/t-km, accounts for the high variation in the payload and distance travelled. The baseline CO_2_ emissions (g/km) show greater variation, and the fuel consumption values across the different truck subgroups vary between 24 L/100 km and 33 L/100 km [5].

A reduction in vehicle drag will most probably contribute to fuel saving and decreased air pollution, probably almost independent of vehicle purpose [6]. This possibility encouraged us to study truck streamlining more closely.

In nature, many organisms have adapted to aquatic or aerial life, which involves fast swimming or flying, and their streamlined shape has been adapted and optimized over millions of years [7,8,9]. This provides us with a catalog of possible ideas and solutions to pick from as role models. Next to that, undertaking ‘industrial espionage’ on biological role models (e.g., trout, shark, dolphin, etc.) can potentially save a serious amount of development time because it provides us with a workable starting point that only has to be translated to human technology. Although it may be difficult to see at first glance, a fish swimming in water has some similarities to a truck moving through air. Due to its streamlined shape, a trout reduces its drag force significantly, and it is not hard to imagine that its drag coefficient is much lower than that of the cubic box-like shape of a truck (see Figure 1a). In fact, the ratio between the two is close to 10 [10,11,12,13,14].

This encouraged us to follow a biomimetic optimization procedure for our truck streamlining study. We will use a formalized method to analyze the outline of a biological model. We derive a series of organism-inspired designs that will be tested using 3D computational fluid dynamics (CFD). In that way, we can use the elements or characteristics of biological streamlining to redesign the shape of the truck. Of course, we aim to improve the characteristics of the truck’s design with respect to drag.

The resistance of a truck moving through the air is composed of two parts: skin friction drag and pressure drag [14] (see Figure 1b). In this study, we concentrate on reducing pressure drag due to the fact that skin friction drag hardly varies in relation to the shape of the truck [15]. The air resistance increases in importance at increased vehicle speed. Skin friction drag hardly changes during such speed changes, but pressure drag increases dramatically with speed [15,16,17]. Therefore, when aiming to reduce fuel use, a reduction in pressure drag seems to be the most logical choice, with the highest likelihood of resulting in emission reductions.

Thus, many manufacturers of heavy commercial vehicles as well as energy research institutes, aim to increase transport efficiency by improving the truck’s aerodynamic design to improve engine fuel efficiency [16]. In this paper, we present a novel truck aerodynamic design procedure for a truck shape that significantly reduces drag and thereby fuel consumption and exhaust emissions. At the same time, it should fit well within EU regulations and maintain the truck’s normal use and stability. Complementary to many studies so far, we plan to concentrate on reshaping the outline of the truck’s tractor and not the trailer [2,3,5,16,18]. Our new tractor design will be inspired by biological shapes with a known high level of streamlining and low drag, e.g., the head of a trout, but also other model organisms. In a preliminary study with a rather coarse biomimetic modelling procedure, we achieved measured drag reductions of up to 40% at cruising speed, potentially resulting in fuel consumption reductions of up to 20% at cruising speed [19].

In the Methods, as well as in the Discussion section, the biological model selection procedure, the design method using computer-aided design (CAD), the performance analysis using CFD and the planned experimental follow-ups are explained and evaluated. There, the unique points of a biomimetic pathway, as applied in this study, are indicated. The selection process ultimately assists us in finding the best-performing model(s) for the experimental work in follow-up studies. Such studies will not only allow us to verify the results of our CFD but can also provide experimental evidence supporting the present study. In the conclusion section, we summarize our findings with regard to the bio-inspired designs, the design method, the CFD simulations and the aerodynamic performance.

## 2. Materials and Methods

### 2.1. Models

(i)Rationale

A certain standardized pathway for the biomimetic design process was applied (see Figure 2). The first step in this process relates to the selection of the biological model, but the question of ‘how to translate the biological shape to a technical variant’ already plays a role here. A streamlined design for large vehicles can, in principle, be inspired by a range of biological models because streamlining can be found in many animal taxa. Therefore, potential similarities between a cruising road vehicle and a streamlined organism have to be evaluated to discover which animals might be used for our biomimetic design process. First, all streamlined animals are sorted into two main groups: flying in air and swimming in water [8,20,21]. Subsequently, several evaluation rules are used to sort the proper biological models based on the available data, which will eventually result in a final methodology for the bio-inspired process.

(ii)Streamlined animal

In nature, many animals have streamlined shapes adapted to their living environment [7,8,9]. Such animals can live in air, water, or in both. Aerial animals cannot be used as a biomimetic model for our study because, in most cases, the beak is too prominent and only the body is streamlined [8,22,23,24,25,26]. Aquatic animals, particularly fish, are favored for our biomimetic study due to their often all-body streamlined shapes, including the beak/mouth. Fish can be characterized by the habitat they live in, grouping them as follows: pelagic fish, tuna; benthic fish, flounder; and demersal fish, trout [7,26,27,28,29].

(iii)Model selection method

Aquatic animals living near the seabed (so-called demersal swimmers) show a high level of similarity to a truck driving over a road. In both cases, the object moves rather fast but close to a boundary below. Organisms that swim close to the bottom boundary can be assumed to have undergone an evolutionary optimization process that makes them suitable for such physical circumstances [23,24]. This fact skews the selection process toward the group of demersal swimmers, e.g., fish swimming close to the seabed or swimming close to a riverbed.

We tried to find a species that has a certain fluid mechanic similarity to the truck. We used the Reynolds Number (Re=density×velocity×length/dynamic viscosity) as the similarity parameter, aiming for a Re in the range of one million (10^6^). Comparable Reynolds numbers result in comparable fluid flow conditions.

In the real world, the truck is moving relative to static air and a static road. A fish is moving relative to static water and a static seabed or relative to streaming water and a riverbed but keeping some distance between the body and the bottom. Therefore, the formalization/optimization process should test the designs under a similar relative motion. For example, a simulation that uses a static model truck should be performed with a static model and fluid passing by [30].

In summary, the proper biological models will be selected and optimized via an evaluation based on the four similarities described below, which effectively come down to geometrical as well as Reynolds number similarity:(i)the medium is fluid;(ii)a gap exists between the object and the lower boundary;(iii)similar kinetic conditions;(iv)relative motion similarity.

### 2.2. Species Selection

Based on the rules set out in Section 2.1, rainbow trout (*Oncorhynchus mykiss*) was selected to be the first suitable biological model for our biomimetic pathway because it follows all of the criteria. Although there are definitely more demersal swimmers that follow the criteria, we decided to start with trout because our research group has substantial experience with trout. Adult stream-dwelling (fluvial) rainbow trout usually grow to between 30 and 50 cm in length and grow to a body mass between 0.5 and 2 kg, depending on various factors mostly related to habitat and genetics [31]. The estimated overall average swimming speed for rainbow trout is around 0.84 m/s (SE = 0.02), with a 95% confidence interval of 0.79–0.89 m/s [27]. Rainbow trout can reach speeds as high as 2.72 m/s.

The Reynolds number of trout with a potential top speed of 2.72 m/s is calculated with the following formula:$$Re=\frac{\rho ul}{\mu }=1000\;\mathrm{kg}/m^{3}\times 2.72\;m/s\times \frac{0.40\;m}{8.90\times 10^{-4}\;\mathrm{Pa}\cdot s}=1.2\times 10^{6}$$
where *Re* represents the Reynolds number, ρ is the density of the water, u is the velocity of the trout, l is the body length of the trout and μ is the dynamical viscosity of the water, respectively.

For comparison, the Reynolds number of a truck on a highway at cruising speed is calculated with the following formula:$$Re=\frac{\rho_{air}u_{truck}l_{truck}}{\mu_{air}}=1.29\;\mathrm{kg}/m^{3}\times 25\;m/s\times \frac{2.25\;m}{1.81\times 10^{-5}\;\mathrm{Pa}\cdot s}=4.0\times 10^{6}$$
where *Re* represents the Reynolds number, $\rho_{air}$ is the density of the air, $u_{truck}$ is the velocity of the truck (25 m/s = 90 km/h, based on EU highway speed regulations [32,33]), $l_{truck}$ is the length of the tractor and $\mu_{air}$ is the dynamical viscosity of the air, respectively. The calculations were performed with standard atmosphere and room temperature values.

From the above calculations, we can see that the trout has a similar but slightly lower Reynolds number compared to the truck; therefore, trout fit our criteria (i) to (iv) well enough to serve as the model for a streamlined truck design.

### 2.3. Formalization

#### 2.3.1. EU Standards

Recent, EU regulations relating to the size constraints of an articulated truck–trailer configuration to allow for design changes aimed at reducing fuel consumption and emission levels [16,34,35]. The new regulations allow for a length increase, which should not be designated for increased payload volume but to allow for design changes that result in decreased fuel usage and lower emissions. At the same time, the truck + trailer’s road behavior, especially on narrow roads, curves and roundabouts, should be the same as before [19]. Within these boundary conditions, we see many possibilities for serious design changes that will have the desired effects, e.g., no mirrors, extended truck noses, rounder corners, improved driver field of view, etc. As Figure 3 shows, the outer (red radius) and inner (blue radius) circle of a minimal-size roundabout according to EU regulations are, respectively, 12.5 m and 5.3 m. The green region is the possible extension space for the new tractor design [16]. The streamlined design will have to match or be smaller than the round, green-colored extension to stay within EU regulations [16]. This yields about 1.5 m extra length when designed appropriately.

In the following sections, Table 1 will be used as the outer limits of our bio-inspired designs and to maintain the uniformity of all of the geometrical transformations and CFD simulations [34]. The numbers indicated in bold, i.e., the length and the length–width ratio of the new tractor, quantify the design space of any streamlined nose extension (see Figure 3). The biological model is translated to human technology by applying CAD. Furthermore, the resulting designs are, in principle, flexible with regard to further shape changes due to the variable settings in the design process. After testing for a number of different variables, a series of tractor designs will be selected for subsequent CFD analysis, which results in comparative data related to their aerodynamic performance.

#### 2.3.2. Scaling and Curve Fitting Method

The design outline of the truck is inspired by the biological model. Because the actual dimensions or even actual dimension ratios of the redesign will not be the same as the biological model, a reshaping process is necessary for the formalization and potential application of the animal morphology. The shape of the head of the trout from the top view and lateral view is formalized to derive curvature data that obey our criteria extrapolated from the EU regulations. In this study, we use a digital camera to determine the outline of the trout. Furthermore, a deforming foreground grid is used to reshape the head of the tout toward the desired dimension ratio based on data in Table 1 (see Figure 4 and Figure 5).

The rainbow trout’s body is meshed with a series of equidistant vertical lines that divide the body of the trout into sections with equal widths (Figure 4 and Figure 5). Subsequently, a deformation process is applied to the part of the trout that needs to be re-scaled, in this case, the front part from where the outline slope is horizontal (just in front of the dorsal fin) to where the outline slope is vertical (just above the mouth opening) (see Figure 4 and Figure 5). In this process, the lines in the deforming part are kept equidistant but at varying distances until the necessary re-scaling has been accomplished. Then, the morphometric information of the deformed biological model is analyzed and further processed into a smooth curve. The result is, in this case, a curve following the outline of the trout body in the target part but compressed along the X-axis. Finally, the data are fitted from the compressed outline to a mathematical function (ImageJ) that can be further used in the truck design process. In summary:import the appropriately compressed picture of the fish into ‘ImageJ’;manually track the curve and plot the tracked dots (see Figure 6);output the dot coordinates and other characterizing data;use curve fitting to derive a mathematical function.

In Figure 6, the blue circles marked on the head of the trout have been fitted to a Gamma function, as follows in Equation (1):(1)$$y=b{\left(x-a\right)}^{c}e^{-\left(x-a\right)/d}$$

The same procedure is followed for (half of) the top view of the trout’s head (see Figure 5), which results in a corresponding gamma function for the deformed top view outline.

Figure 7 shows the fitted curves for the dorsal view (top view) and the lateral view (side view). The (example) parameters after curve fitting are given below, including their R-squared values (see Table 2):

The two corresponding gamma functions are now used to construct the outline of the streamlined truck design. The side view outline will be constructed from the lateral curve fitting result, and the top view outline will be formed by a mirror flipping of the dorsal curve fitting result (see Figure 8).

### 2.4. CAD Design Step

The final result of the curve fitting process will be input for a computer-aided design (CAD) process, with the two lines representing the outlines of the truck.

To accomplish this, all of the parameter values are imported into a 3D coordinate system: *X*, *Y* and *Z*. The *X*–*Z* plane represents the side view, and the *X*–*Y* plane represents the top view. Then, the corresponding functions are as follows:

Side view:(2)$$f_{side}\overset{\mathrm{def}}{=}z=f\left(x\right)=9.48\cdot {\left(x-31.7\right)}^{0.57}e^{-\left(x-31.7\right)/453.837}$$

Dorsal view (for +X and −X to represent the two halves):(3)$$f_{top}=f_{t}\overset{\mathrm{def}}{=}\left\{\begin{array}{c} y=g\left(x\right)=183.467\cdot {\left(x-28.96\right)}^{0.1648}e^{-\left(x-28.96\right)/2969.9}\\ y=g\left(-x\right)=183.467\cdot {\left(-x-28.96\right)}^{0.1648}e^{\left(x+28.96\right)/2969.9} \end{array}\right.$$

These curves are visible in a 3D plot in Figure 8 as a blue curve (side view outline, Equation (4)) and a ‘left’ and ‘right’ red curve, together forming the top view (Equations (5) and (6)).
(4)$$z_{blue}=f\left(x\right)-f\left(x_{0}\right)$$
(5)$$y_{red1}=g\left(x\right)-g\left(x_{0}\right)$$
(6)$$y_{red2}=-g\left(x\right)+g\left(x_{0}\right)$$

The two curves of the side view and top view are now expanded to a 3D volume by scaling the side view curve according to its position on the bottom curve. Figure 9a shows the central side view curve as well as two intermediate curves to the left and to the right. Figure 9b shows a whole series of curves on one side of the central curve, indicating one side of the 3D volume. Note that the rear side of the 3D volume is rectangular because the front side of a trailer is rectangular; the height–width ratio (H/W) of the rectangle is 1.35 (see Table 1), similar to the trailer.

Following the above-described procedure directly results in a trout-shape-derived model named Model O. This model will also serve as the starting point for further design models. From here on, Model O can be modified, e.g., by rounding the bottom surface through the manipulation of additional parameters. 

Figure 10 shows a round-nose design derived from Model O in side view after applying Equations (7) and (8). These equations effectively scale the Model O curve to a lower height but add an additional curve below, resulting in a rounded nose.
(7)$$z_{1}=\alpha \cdot z\left(x\right)+\left(1-\alpha \right)\cdot H$$
(8)$$z_{2}=\left(1-\alpha \right)\cdot -z\left(x\right)+\left(1-\alpha \right)\cdot H$$
here $z_{1}$ and $z_{2}$ are functions of the middle plane’s outlines, which can be seen in Figure 10. α is the scaling factor, which varies on an interval [0, 1]. $z\left(x\right)$ is the gamma function from Equation (2). H is the height of the tractor. 

As an example, when α=0.15 in new model A, the middle plane’s outline results from scaling the gamma functions to $0.85z\left(x\right)$ and $-0.15z\left(x\right),$ respectively, in the vertical direction. Figure 11 shows the resulting curves for such a transformation. 

Figure 11b shows the round-nose model from Figure 10 represented as a side view curve. The nose-rounding procedure has been repeated for the scaling factors of 0.15, 0.25, 0.40 and 0.50 yielding four different round-nose models which will be tested using CFD simulations.

### 2.5. Computational Fluid Dynamics (CFD) Simulations

In this study, the essence of CFD is carrying out simulations in a virtual wind tunnel for all of the bio-inspired truck designs by using numerical calculation methods. CFD can simulate the flow patterns around any shape and thereby give an impression of the fluid dynamic performance of those shapes, in this case, a series of truck designs. Thereby, the truck models can be characterized by calculating their drag forces and evaluated and compared to one another as well as to real truck designs to quantify if emission reductions will indeed be accomplished. All of the CFD simulations have been performed using ‘COMSOL Multiphysics 5.5’ [36].

#### 2.5.1. Geometry and Materials

The CAD designs of a series of derived 3D shapes resulting from Section 2.4 have been visually compared, and five characteristically different designs standing for five bio-inspired truck designs have been selected for testing with CFD simulations. These five designs were imported into COMSOL and transformed into solid objects and introduced (one at a time) in a virtual/simulated wind tunnel. To simulate the road underneath the truck, a flow splitter is introduced upstream of the truck cockpit. The truck design is positioned just above the plane of the flow splitter with a suspension gap (SG) in between (see Figure 12) (NB: mind you, the wheels are left out of all simulations). Here, the splitter could be replaced by a moving conveyer-belt-like floor, but for reasons of comparability to other experimental work, we applied a static splitter plate. The Results section also contains some data related to the models without a splitter plate for the sake of comparison.

The splitter plate should extend at least two times the length of the object upstream and five times its length downstream [37]. With an L = 8.5 m truck, the upstream length is minimally 2 L and the downstream length is minimally 5 L. The total domain length being minimally 8 L (see Figure 13) [37]. In our CFD work, we chose a total length of 75 m (see Table 3). The length of the trailer was taken as 5 m, which is relatively shorter than a normal truck. Because we concentrated on the tractor, a shorter trailer received less simulation time. The tractor–trailer gap is set to 0.5 m, again for reasons of reducing simulation time.

In order to confirm the simulation volume was sufficiently large to avoid artifacts, one simulation was completed where the volume was doubled, and which yielded the same results as the original volume. 

#### 2.5.2. Physics

##### Discretization Methods

For the CFD analysis, incompressible flow was simulated with a Reynolds-averaged Navier–Stokes (RANS) turbulence model at k-ε settings [38,39,40,41]. The initial condition is static (velocity = zero), with a uniform pressure comparing the inlet with the outlet. For comparison, a k-ω model was run in parallel, because some literature sources also use a k-ω model for large vehicle analysis [42,43,44]. For our simulation, the k-ε model gave more reliable and more detailed results with regard to the aerodynamics.

##### Boundary Conditions, Inlet, Outlet and Turbulence Intensities

In COMSOL, the surfaces of the truck are non-slip, which simulates skin friction between the air and the truck. The inner surfaces of the virtual wind tunnel are set to slip walls, avoiding boundary layer effects and reducing the calculation time. The top surface of the air splitter is set as a sliding wall moving similar to a conveyor belt at the same speed as the incoming airflow [42]. On the boundary, the turbulent intensity is 0.01 and the turbulence length is 0.1 m [36,42,45], similar to Garry (1996) for a bluff body simulation [46].

All simulations have air as the medium, the inlet oncoming flow velocity was set to 18.5 m/s (66.6 km/h), with all simulation parameters and settings as given in Table 4 [36,42,45]. The outlet setting is under uniform pressure compared to the initial conditions and with suppressed backflow. In a follow-up study all simulation outcomes will be verified experimentally during a wind tunnel study [32,47,48].

#### 2.5.3. Mesh Study

A ‘normal’ mesh was applied in our simulations. The type of mesh used was triangle elements on the surface of the truck and for the boundaries, and a tetrahedral mesh in the space between the truck and the boundaries (see Figure 14a). These mesh settings have been used in several other studies to simulate a vehicle rolling over a ground surface [38,39,40,41,43,44,49]. To test for mesh coarseness, simulations were performed using two different element sizes, comparing the truck’s total drag force and the net drag coefficient. Figure 14b shows a ‘normal’ mesh with element sizes being minimally 0.000395 m and maximally 0.09125 m; the relative coarser mesh has elements ranging in size between 0.00058 m and 0.13615 m. The simulations with varying mesh coarseness to test grid dependency with respect to the calculated drag coefficient of the truck were performed on Model A [44,49]. Results of the grid dependency test appeared to be almost similar for all three mesh element sizes of 0.363 (fine mesh), 0.364 (normal mesh) and 0.372 (coarse mesh), with ‘coarse’ being a slightly coarser mesh than ‘normal’ and ‘fine’ being a slightly finer mesh than ‘normal’ [42]. The number of elements for four mesh sizes are shown in Table A3 (Appendix). The number of layers simulated in the boundary layer is 8. The Boundary layer stretching factor is 1.2. (here 1.2 means that the thickness increases by 20% from one layer to the next). The thickness of the first element layer was defined as 1/20 of the local domain element height [42,45] (see Figure 14c). Increasing the number of layers to 12 in ‘normal’ mesh or 16 in ‘fine’ mesh did not significantly change the simulation results (see Table A4). We used 8 layers in the boundary layer to maintain efficient calculation time. All simulations were solved for y+ values in the range of 30 - 800 (k-ε turbulence model); the y+ distribution graph of Model A is shown in Figure 14d, indicating the minimum value is 39.58. Based on these results, the ‘normal’ mesh size is chosen because it is sufficiently accurate for all of the simulation cases but requires less calculation time.

#### 2.5.4. Tractor Outline, Trailer, Nose Rounding

In this study, we looked at the effect of the direct derivation of a tractor outline from the trout bio-model. However, we were also looking at the additional effect of a rounded nose versus a sharp-edged nose. For the rounded nose models, a rounding was superimposed on the sharp-edged nose model, the rounding being a quarter circle with the radius being 15, 25, 40 and 50% of the truck cabin height, respectively.

#### 2.5.5. Solver Algorithms and Iterative Convergence Criteria

Since stationary simulations were performed in all cases (uniform oncoming velocity without changes in time), the derived solutions were assumed to be time-independent. The linear system solver uses the restarted GMRES (generalized minimum residual) method. This is an iterative method for general linear systems of the form Ax = b. By using a discretization method, there is a segregated solver that iterates the velocity, pressure and turbulence variables until a 10^−5^ deviation level is reached [42].

#### 2.5.6. Parameter Definition

The total drag force was calculated by using a surface integration for the tractor and/or trailer. The net force acting on the direction of oncoming flow is the drag force, which is the sum of pressure forces and skin friction forces, and any resulting vertical forces were indicated as downforce, which potentially contributes to the stability of the truck [42]. The drag coefficient of Cd is derived from the drag force and the truck’s geometrical information and was calculated for comparison reasons.

The drag force is defined as indicated in Equation (9), and the derived drag coefficient as indicated in Equation (10):(9)$$F_{drag}=\frac{1}{2}C_{d}\rho Au^{2}$$
(10)$$C_{d}=2F_{drag}/\left(\rho Au^{2}\right)$$

The coefficient of pressure indicates the scale of the total pressure on the surface of the truck. Here, the coefficient of pressure is defined in Equation (11) as follows:(11)$$C_{p}=2\left(p-p_{\infty }\right)/\left(\rho_{\infty }u_{\infty }^{2}\right)$$
where A is the cross-section area of the truck, $p_{\infty }$ is the static pressure in the freestream and $u_{\infty }$ is the freestream velocity of the fluid or the velocity of the body through the fluid [50,51,52].

#### 2.5.7. CFD Validation

The grid convergence method (GCM) is an accepted and a recommended method that is based on the Richardson extrapolation in CFD studies [53,54,55]. The grid convergence index (GCI^21^_normal_) values of several locations in the Model A simulation at the front of the truck model have been evaluated and found to never to surpass 2.02% (Table A1).

A validation and error analysis can be given here based on the results of a comparison of CFD with the experimental results of a truck model tested and simulated at the same Re as have been indicated in this paper. The difference is 5.36% maximally for the Original model and 5.56% in Model design B (see Figure A1). We regard this as a good match. The physics, mesh and boundary setting also have been validated in several similar studies [40,41,43,44,49]. In addition, the simulation results of different turbulence models, mesh sizes and boundary layers have also been validated by varying the CFD simulations settings (see Table A4).

Our study was principally similar to other truck aerodynamics studies using CFD at varying CFD-parameters applied at Re = 10^6^ [38,39,40,41,43,44,49,53,54]. Our CFD study may additionally indicate potential benefits from the bio-inspired design process. In a future study, we plan to also experimentally analyze the original and the optimized truck models to compare to flow tank experiments as well as a wind tunnel study [6,32,47,48,56,57,58] varying similar parameters as in this study.

## 3. Results

### 3.1. Bio-Inspired Designs

#### 3.1.1. Original Model

The original truck model is a simplified version of the DAF XF105 tractor, as can be seen in Figure 15. This somewhat simplified version maintains the aerodynamic characteristics of the tractor but omits aerodynamically non-important structures.

#### 3.1.2. Sharp-Nose Model

In Figure 15, in the top-middle panel is the design model O as the first output from CAD and is directly derived from the two curves (lateral outline and dorsal outline) of the trout, as indicated in Section 2.4.

#### 3.1.3. Round-Nose Model

Four different rounded-nose configurations were tested to examine the effect of a more round-nosed design. In all cases, the top view outline was left unchanged with respect to Model O, but in the side view, the rounding off of the sharp lower front edge of Model O changed the appearance and, potentially, the aerodynamics (Figure 15). The models indicated as A, B, C and D have gradual nose rounding with the apex, respectively, at 15, 25, 40 and 50% of the cabin height.

### 3.2. CFD Results

Figure 16 shows an example picture of the CFD results with regard to the air velocity in side view (lateral) as well as in top view (dorsal); the local air velocity can be derived from the color coding (see color bar legend). The planes of visualization were located horizontally through the nose stagnation point which is around 30% of the height of the truck, and vertically through the middle plane.

The resulting values for the drag and the drag coefficient (Equation (10)) for an oncoming airflow of 18.5 m/s show that all models (O and A–D) have significantly reduced drag when compared to the original truck model, as can be seen in Table 5 and Figure 16a. These are drag values for the tractor, including the trailer. Striking is that all drag coefficient values are reduced between 33 and 42.5% in comparison to the original truck design, but the designs O and A–D do not seem to differ that much from one another. All of the drag forces were calculated by surface integration, including pressure force and friction force [42].

A comparison of the drag values for the tractor excluding the trailer to the tractor including the trailer can be seen in Table 6 and Figure 17b for the normal truck model (top) and the Models O and A–D. The rounded nose models again show an even further reduced drag coefficient, up to 40.7% (model B). There seems to be little difference between the rounded nose models, although there appears to be a tendency that a rounding low on the nose gives a lower drag compared to a rounding higher up the nose (comparing models C and D). When looking at differences between the tractor only vs. tractor + trailer, it seems that a trailer might not increase the drag of the truck but might actually mitigate drag effects (see Table 5 and Table 6). The CFD results (in 18.5 m/s airflow) show the Cd of the tractor + trailer (see Table 5) to be somewhat lower compared to the tractor only (see Table 6) but still at a comparable value. Although it may be too early to draw conclusions, it may be caused by the quite sudden flow changes immediately behind the tractor (tractor only, Figure 17b), whereas these sudden changes may be mitigated by the presence of the trailer (see Figure 17a).

The drag values for the tractor including the trailer compared to the tractor without the trailer, are shown in Table 7 and Table 8. The splitter plate was removed, and the bottom surface boundary condition was replaced by a sliding wall with the same velocity as the oncoming airflow (18.5 m/s) (see Figure 18). The drag coefficients show a drag reduction in Model O-D. Comparing Table 5 and Table 7 and Table 6 and Table 8, the splitter hardly changes the drag force and drag coefficient of the truck with or without a trailer.

The pressure distribution and the pressure coefficient along the centerline of the top and bottom surfaces of Model A are shown in Figure 19 (for a 3D impression of the pressure distribution see Appendix A). The flow pattern and consequent pressure distribution around the tractor and the trailer were simulated with the only a gap distance between the truck and the lower wall, which was set as a sliding wall with a wall speed similar to the oncoming wind speed [42]. The pressure contour middle slice shows a higher pressure at the front part of the tractor and relatively low pressures all over the truck–trailer combination, particularly in the tractor–trailer gap and behind the trailer. The blue curves in Figure 19b show the pressure coefficient along the top surface of the combination (upper graph) as well as along the bottom surface (lower graph). The pressure on the top surface of the tractor decreases rapidly from the X/L value of 0 to 0.23 and increases again slightly towards the end (X/L = 1, X is the distance along the combination starting at the very front, L is the length of the combination). The pressure coefficient curve for the bottom surface shows a steep to strong negative values even before X/L = 0.05 and then gradually increases again. The wavy region at X/L = 0.38–0.42 shows the effect of the gap between the tractor and trailer.

In this study, we also looked at the relative importance of skin friction drag (see Table 9). All of the friction forces were calculated by a surface integration, including wall function calculation [42]. The skin friction of design models O and A–D stays pretty stable relative to the original truck, but when expressing it as a fraction of the total resistance, it is relatively somewhat larger. The reason for this result is the reduced pressure drag and little change in frictional resistance.

The airflow velocity vectors show the magnitude and direction of the air flow passing by the tractor (see Figure 20). For the original truck, several stagnation points are shown in Figure 20. The stagnation points for the design models O and A-D change according to the nose height.

Finally, we compared the axial velocity profiles for all models with three kinds of meshes. Figure 21a (data taken from Model A) presents an axial velocity profile along the z-axis at an axial location of 1.88 L (distance to the inlet) which is at the front of the truck model. The three sets of grids had 739,126, 242,801 and 55,714 cells, respectively. The local order of accuracy p ranges from 1.07 to 8.44, with a global average p_ave_ of 4.37, which is an indication of appropriate accuracy [53,54]. The GCI ranges from 0.004% to 2.3% for 20 different locations along the vertical line in the middle plane just in front of the truck model and are plotted as error bars in Figure 21b. The maximum discretization uncertainty is 2.3%, which corresponds to ±0.36 m/s (see Table A2).

## 4. Discussion

### 4.1. Model Selection Procedure

As indicated in the Section 2, many groups of animals are potentially suitable for bio-inspired streamlining design, but so-called demersal fish (fish that live close to the sea floor) are the preferred group. This is mostly because they swim close to the seafloor, which shows similarity to a truck driving over a road. Other species that might be explored in follow-up studies for this purpose might be demersal sharks. In our model organism selection procedure, our predefined similarity parameters were useful and will also be applied in future model selections: medium, gap, kinetics and conformity. Using trout as a model provided significant drag reduction results, and, in that respect, this model was a successful choice. It is difficult to estimate if other model species might result in even higher reductions; more analyses will be necessary first. 

### 4.2. Design Methodology

The design method indicates a formal process that starts with the selected animal and ends with a truck showing streamlining similarities with respect to the model organism. The outlines of, in this case, a trout were formalized, abstracted and projected to a truck. Additionally, four variants with respect to nose-rounding were designed and analyzed, showing that nose-rounding played role in further reducing drag. Although the original model organism outline had to be morphed to fit the truck design space, the streamlining characteristics are kept to a certain extent and can clearly provide advantages when applied to (in our case) redesigning a truck for drag reduction purposes. 

### 4.3. Computational Fluid Dynamics

The approach of using CFD as a virtual instrument to analyze and verify the performance of our truck models appeared to be very suitable and successful. CFD in itself is a wonderful analysis tool but not easy to apply because it assumes that the users have thorough fluid mechanics and numeric knowledge when it comes to selecting the appropriate analysis method and the turbulence model, etc. The results can easily differ by a factor of two or three when not applying the right turbulence model. In our case, we performed an initial verification with a preliminary test in our flow tank facility and were glad to be able to conclude that the experimental test results were very similar to the numerical CFD results, supporting our conclusions. 

To our surprise, the implementation of a road surface that has the same speed as the incoming airflow instead of a stationary road surface hardly affected the truck drag numbers. This, in a way, shows that future analyses can be performed on stationary road surfaces without over- of under-estimating truck drag. This might again save calculation power/time.

There are still a lot of additional factors and settings of those factors that can be analyzed, such as the trailer size and shape, the addition of wheels, suspension gap, tractor–trailer distance, etc. The simulation method also has a promising potential advantage compared to similar studies at similar Reynolds numbers, and which have a similar rigid body–fluid interaction with a stationary state, for which forces can be analyzed by integrating pressure over the surface area. 

In our study, we analyzed the streamlining performance both in air (as the original truck) and in water, keeping Reynolds number similarity. The results from the simulations in water have not been reported here to avoid data doubling and confusion, but they effectively confirmed the results from analyses in air, and are a promising stepping stone for our follow-up experimental work with physical models in a flow tank. 

The CFD software we used, COMSOL Multiphysics, appeared to be very capable of solving our rigid body–fluid interaction questions. The Reynolds numbers of all CFD simulations in this article are in a range of 10^5^–10^6^. Any further simulations with this range of Reynolds numbers can use the same turbulence model (k-ε) and a similar surface integration method in relation to the calculation of drag, lift, skin friction, etc.

### 4.4. Reduced Drag

The bio-inspired design Model B was found to have the lowest drag compared to the other bio-inspired models and had about half the drag compared to the original truck design. Model B has a moderate level of nose-rounding compared to the other models. We interpret this performance as showing a balance between air going along the sides and the top of the model and air going in between the model and the floor (road) (see also Figure 17 and Figure 18).

### 4.5. Consequences of Drag Reduction in Trucks

This study illustrates that the design of long-distance trucks can be improved dramatically with regard to drag reduction. This drag reduction does result in again quite dramatic fuel use reductions and emission reductions. We found that the drag coefficients of the bio-inspired models are about half the value of the original truck design. Since at cruising velocity, about half of the power of the truck is being used to overcome rolling and other mechanical resistances and about half is being used to overcome aerodynamic drag [16], reducing the drag in the order of magnitude of 50% will result in a potential fuel use reduction of maximally about 25 %, with a similar reduction in emissions. These are significant numbers when considering the number of truck kilometers that are now covered in Europe on a daily basis.

### 4.6. Outlook and Further Research

When considering future steps, the most logical ones for this study are: (1) selecting other animal models and testing if this can change the streamlining performance when the same morphometry and analysis methods are applied; (2) performing flow tank experiments with scale models of (at least) the six models used in this study, and (3) performing wind tunnel experiments with larger scale models (preferably keeping Re-similarity) and test whether the results from the study presented here can also be confirmed in an experimental aerial analysis. 

## 5. Conclusions

Our study allows us to conclude that our selection procedure has effectively helped us find appropriate animals for a biomimetic question related to truck streamlining. The applied design methodology appeared to be an appropriate way to formalize the animal-inspired shape and project that on a truck design while still obeying EU regulations. The front surface of the tractor will become a more streamlined shape, but the exact shape may depend on the model animal selection process. In principle, any variation or replacement will result in a new bio-inspired streamlined design. With regard to the trout-derived design from this study, we witnessed significant improvements with regard to truck drag based on our CFD simulations. Model B, having the second level of nose rounding on top of the first-order trout-based design, had the lowest drag, which we attribute to a balanced air flow along the bottom side of the model compared to the sides and the top. A truck design coming close to this bio-inspired model may result in a reduction in aerodynamic drag close to 45%, which in turn may result in fuel consumption and emission reduction of around 22% at cruising velocity. This shows the bio-model-derived streamlining approach to be potentially very promising and definitely an avenue to be further explored in the quest for fuel use reduction, emission reduction and ultimately, CO_2_ emission reduction, and mitigating greenhouse effects in the Earth’s atmosphere.

## Figures and Tables

**Figure 1 biomimetics-08-00175-f001:**
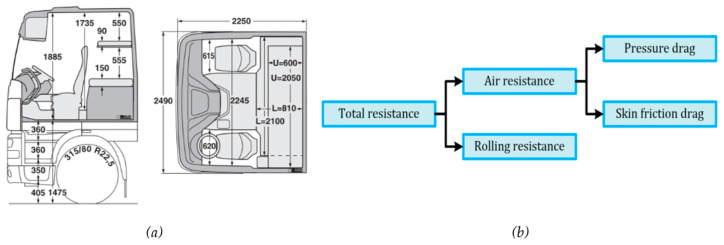
(**a**) DAF XF105 space Cab dimension, (**b**) Force composition.

**Figure 2 biomimetics-08-00175-f002:**
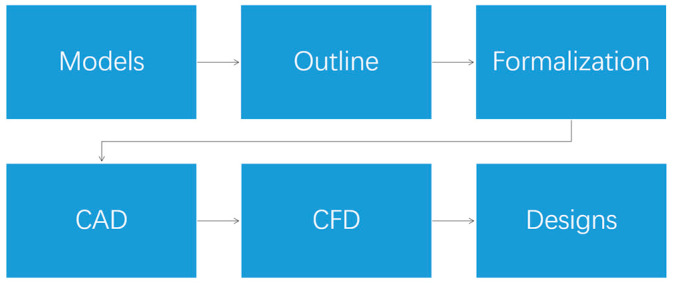
Flowchart of the biomimetic design and post-processing.

**Figure 3 biomimetics-08-00175-f003:**
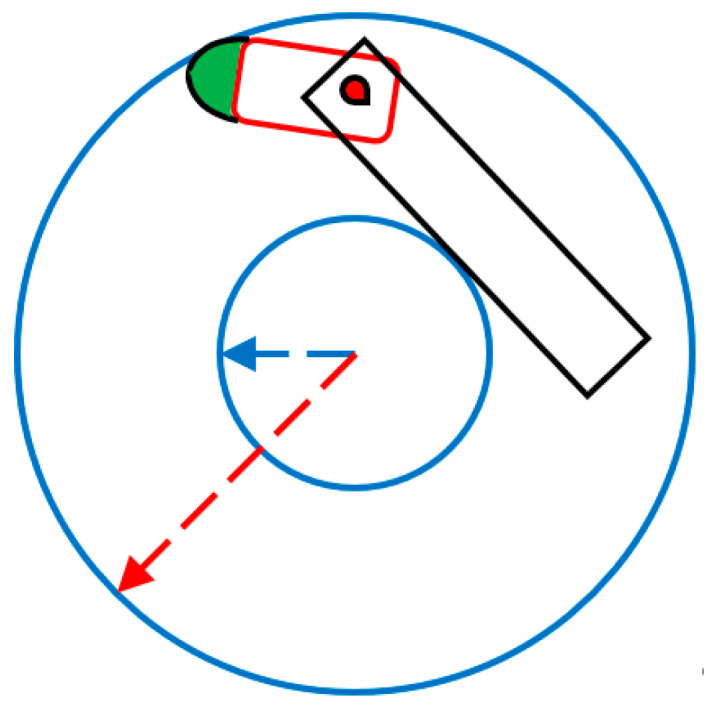
EU transportation vehicle is driving over a small roundabout.

**Figure 4 biomimetics-08-00175-f004:**
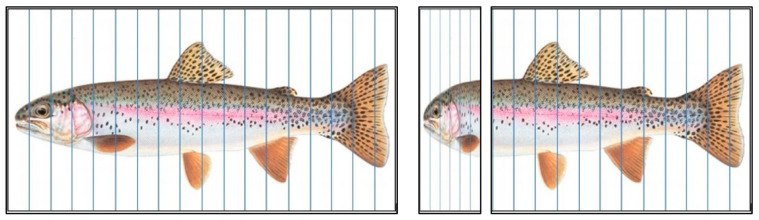
Trout with a deforming grid for the head region, in side view (Scale factor is 0.327).

**Figure 5 biomimetics-08-00175-f005:**
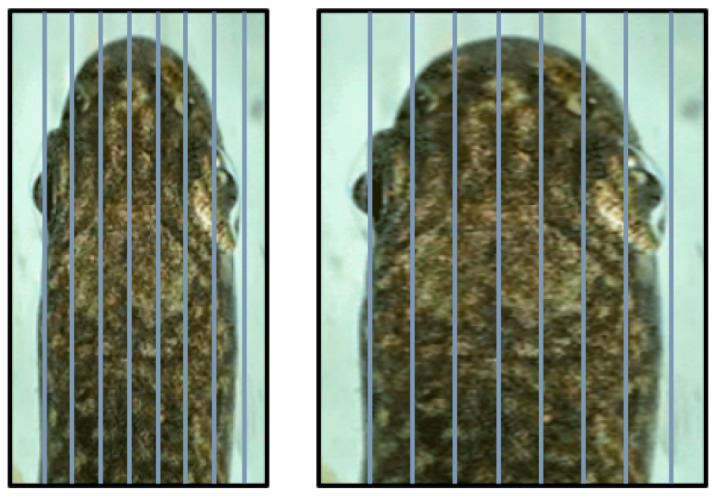
Trout with a deforming grid for the head region, in dorsal view (Scale factor is 1.125).

**Figure 6 biomimetics-08-00175-f006:**
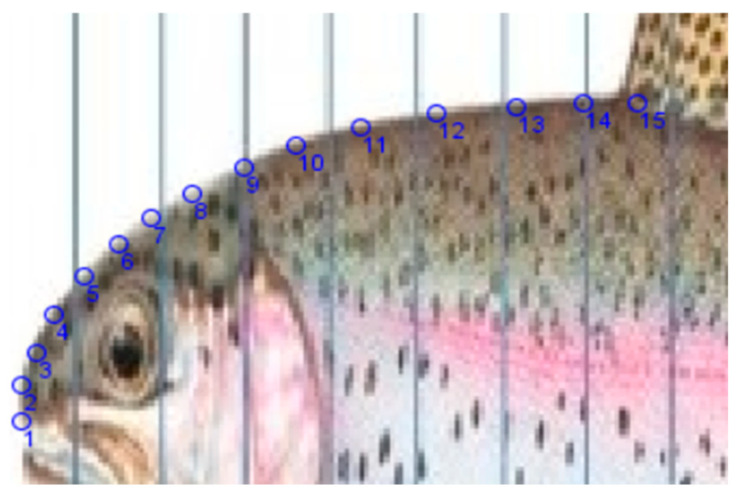
Outline points (Number 1–15) picked up, the side view of the trout’s head.

**Figure 7 biomimetics-08-00175-f007:**
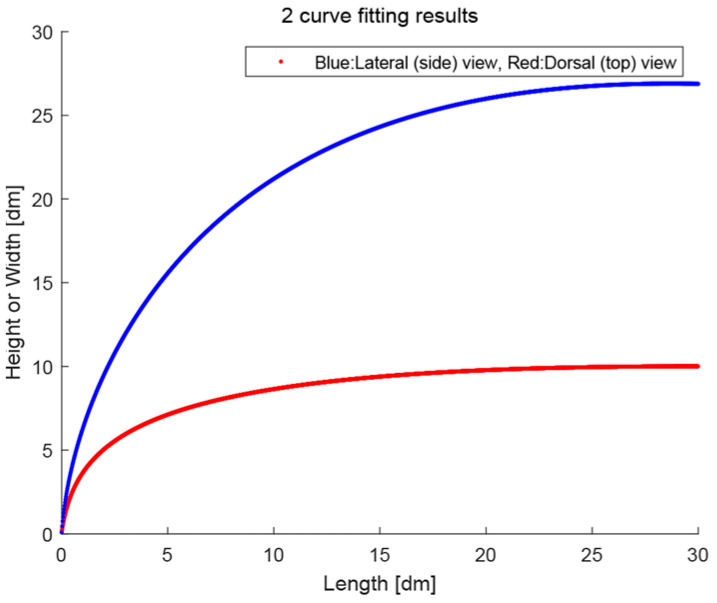
Curve fitting results of the trout-derived measurements after rescaling. The blue curve is based on the side view of the dorsal part of the trout head, and the red curve is based on the top view of the right side of the trout head. The two curves will be used in the design process (see also Figure 8).

**Figure 8 biomimetics-08-00175-f008:**
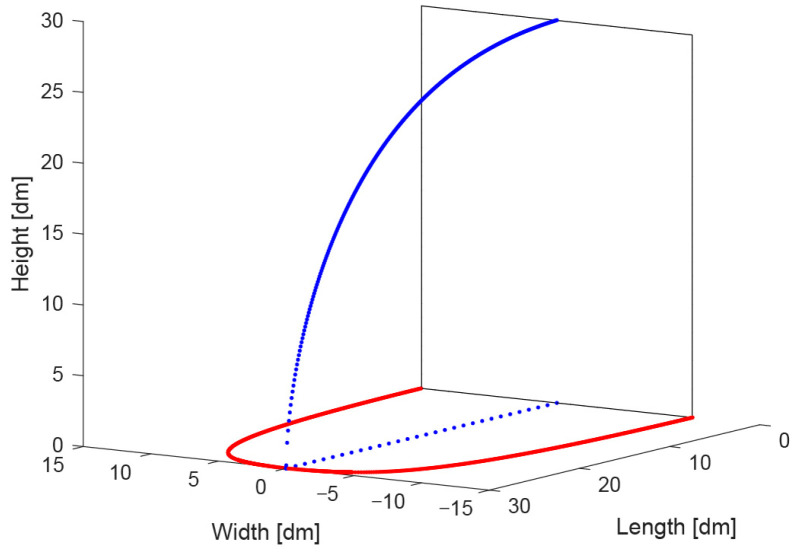
The two curve fitting results and background grids (see text and the caption of Figure 7).

**Figure 9 biomimetics-08-00175-f009:**
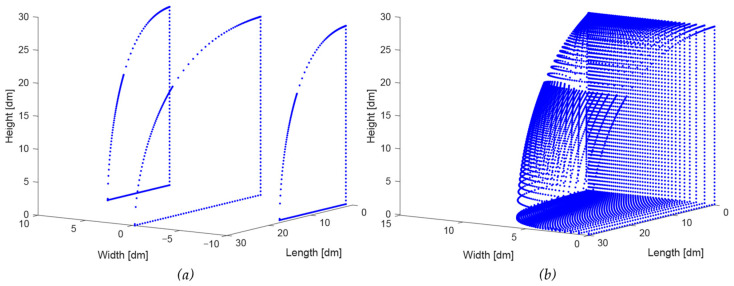
(**a**) The 3 vertical sections of the expansion process, (**b**) a whole series of curves on one side of the central curve.

**Figure 10 biomimetics-08-00175-f010:**
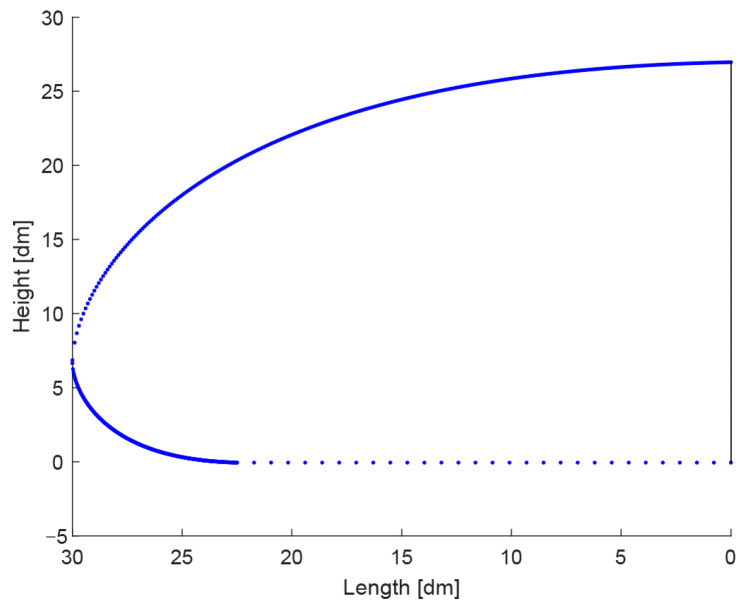
The middle plane of the round-nose designs (compared to sharp-nose designs).

**Figure 11 biomimetics-08-00175-f011:**
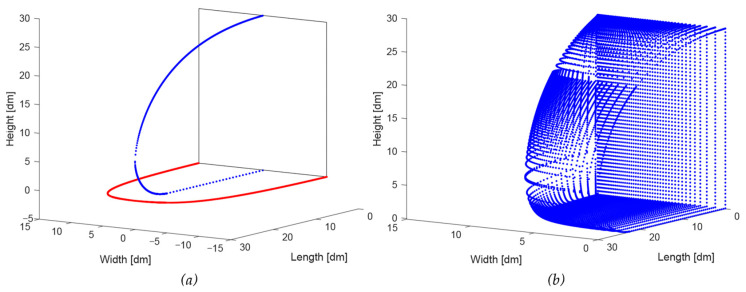
(**a**) The edges and central section of the round-nose design (see also the caption of Figure 7), (**b**) a whole series of curves on one side of the central curve.

**Figure 12 biomimetics-08-00175-f012:**
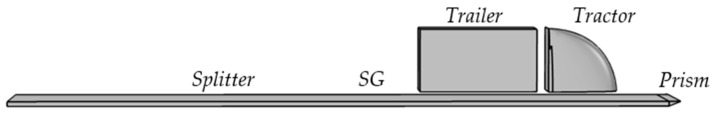
All materials in CFD.

**Figure 13 biomimetics-08-00175-f013:**
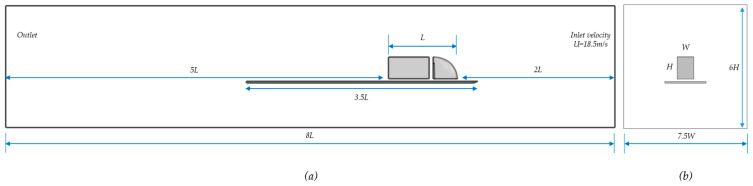
(**a**) Side view, (**b**) Frontal view.

**Figure 14 biomimetics-08-00175-f014:**
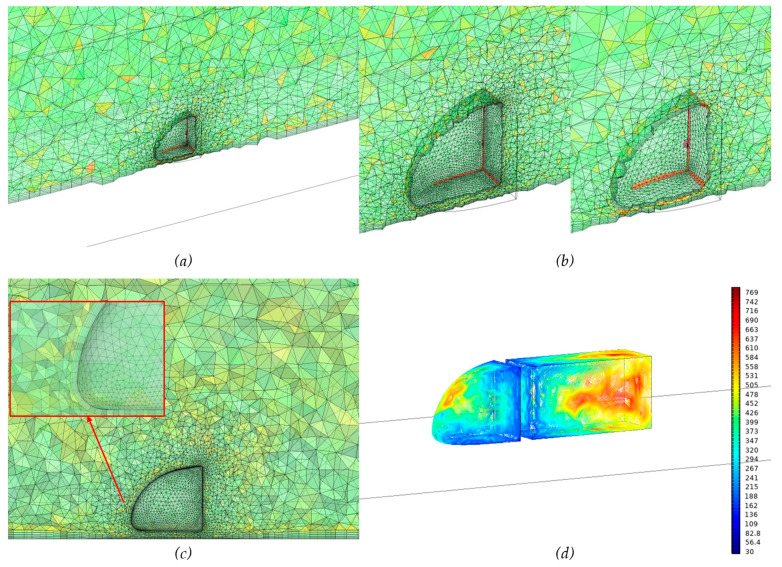
(**a**) Mesh distribution, (**b**) Normal and Coarse mesh, (**c**) Mesh near the Truck, (**d**) y+ distribution at the truck and trailer surface.

**Figure 15 biomimetics-08-00175-f015:**
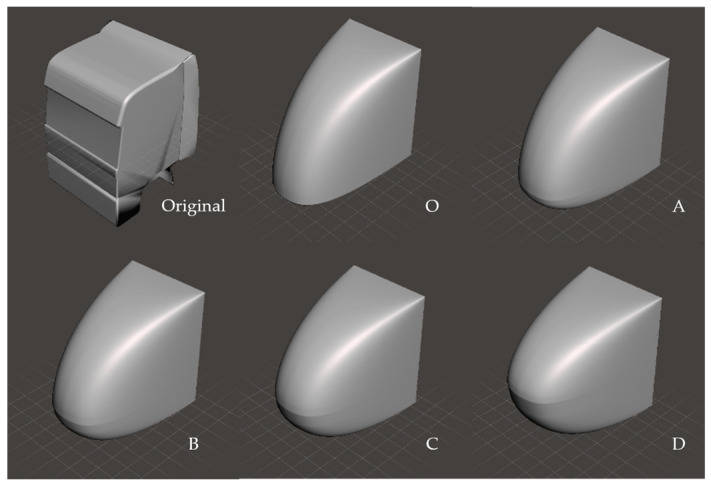
The original simplified model and the optimized/designed models.

**Figure 16 biomimetics-08-00175-f016:**
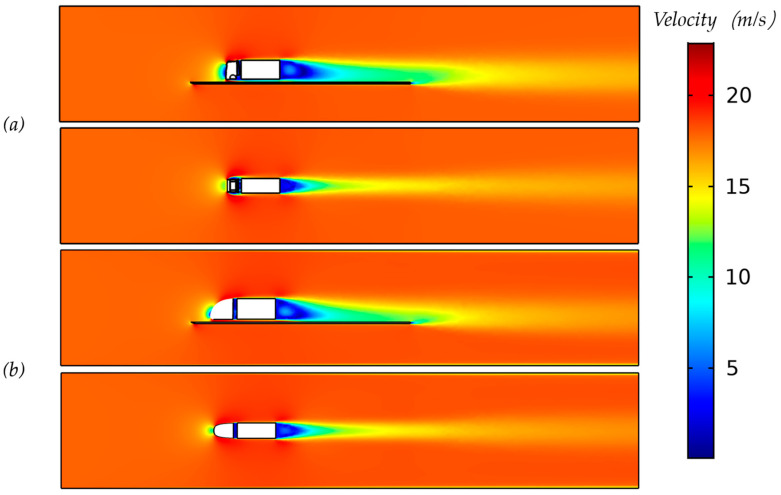
The velocity magnitudes of the original model (**a**) and designed model O (**b**).

**Figure 17 biomimetics-08-00175-f017:**
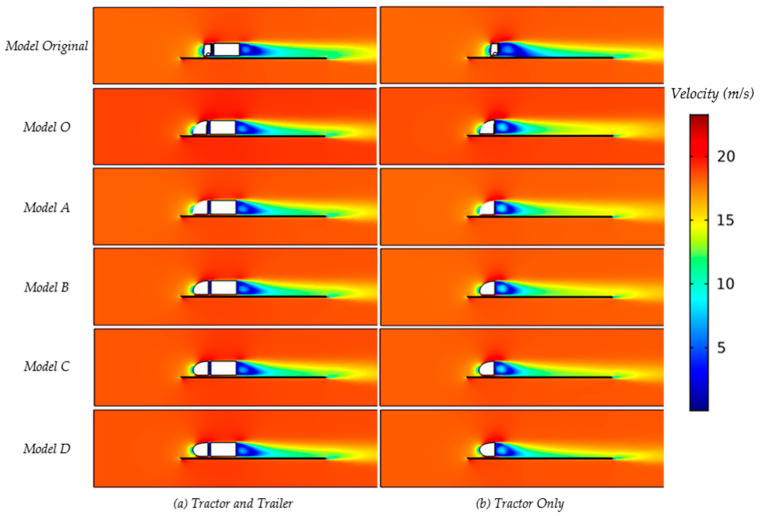
The velocity contour of central slices.

**Figure 18 biomimetics-08-00175-f018:**
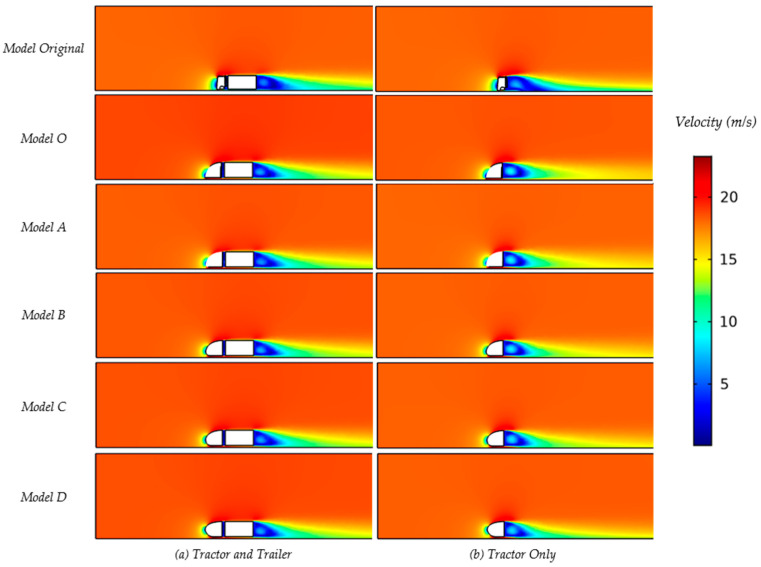
The velocity contour of central slices, no splitter simulation.

**Figure 19 biomimetics-08-00175-f019:**
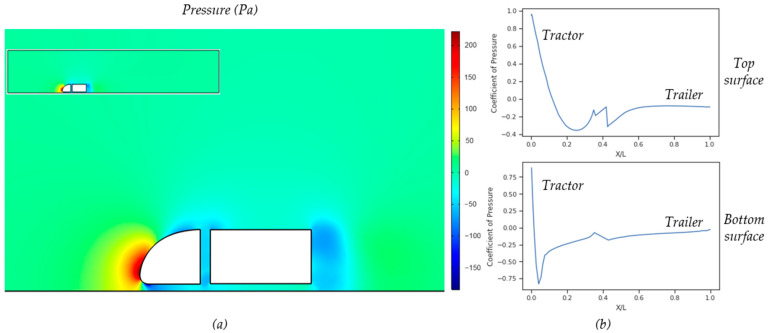
(**a**) Pressure contour, (**b**) Coefficient of pressure alone the centerline.

**Figure 20 biomimetics-08-00175-f020:**
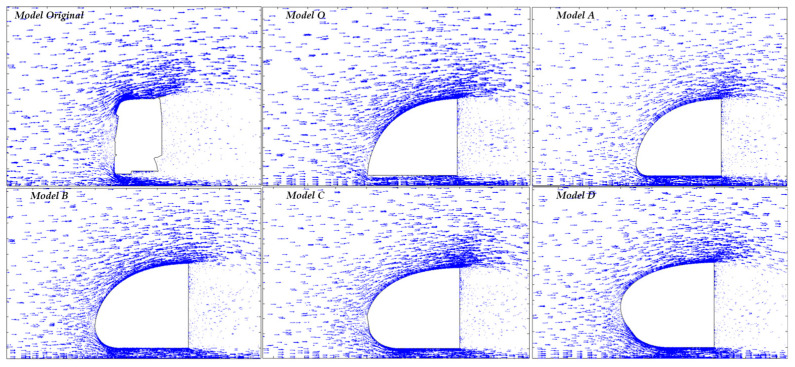
Airflow velocity Vectors for all models.

**Figure 21 biomimetics-08-00175-f021:**
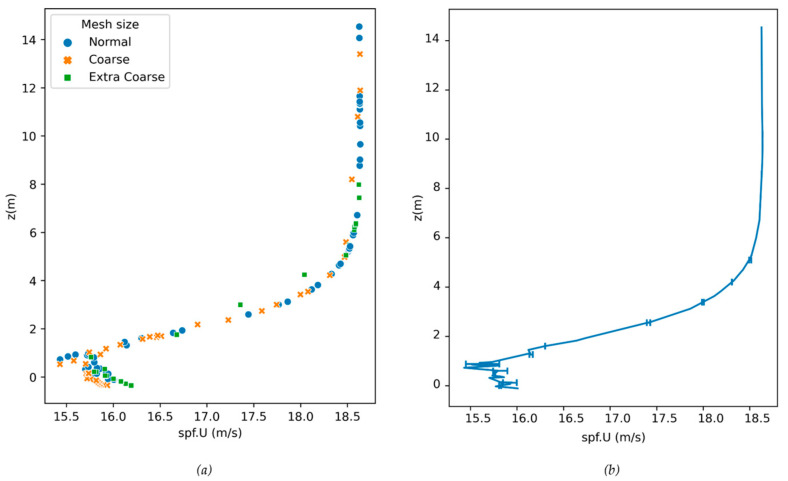
(**a**) Axial velocity (spf.U) profiles for a x, z-plane at the front of the truck model, (**b**) Normal-mesh solution, with discretization error bars by using GCI values.

**Table 1 biomimetics-08-00175-t001:** Redesigned truck’s dimensions calculated by Figure 1a under EU standards [34].

	Original Tractor	Extended Tractor
Height [m]	3.36	3.36
Width [m]	2.49	2.49
H/W	1.35	1.35
Length [m]	2.25	3.75
L/W	0.90	1.5
Wheel Height [m]	1.115	1.115
Tractor and trailer gap [m]	2.25	2.25
Max length of trailer [m]	12	12
Total length [m]	16.5	18

**Table 2 biomimetics-08-00175-t002:** Curve fitting parameters’ data.

Lateral View Curve Fitting Data	Dorsal View Curve Fitting Data
a = 31.7	a = 28.96
b = 9.48	b = 183.467
c = 0.57	c = 0.1648
d = 453.837	d = 2969.9
R^2^ = 0.9993	R^2^ = 0.9987

**Table 3 biomimetics-08-00175-t003:** Dimension of CFD.

Structures and Objects	Length [m]	Width [m]	Height [m]
Tractor	0.35 L	0.24 L	0.31 L
Trailer	0.59 L	0.24 L	0.31 L
Board	3.5 L	0.59 L	0.024 L
Triangular prism	0.059 L	0.59 L	0.024 L
Wind tunnel	8.82 L	1.76 L	1.76 L
Tractor–trailer gap	0.59 L		
Solid Blocking	$\frac{0.2\times 5+2\times 2.65}{15\times 15}$	0.028	

**Table 4 biomimetics-08-00175-t004:** Setting Values of CFD.

Settings	Value
Medium	Air
Inlet	18.5 m/s
Re	3.9 × 10^6^
Turbulent intensity	0.05
Turbulence length scale	0.01
Outlet	Uniform pressure

**Table 5 biomimetics-08-00175-t005:** Model shapes vs. drag coefficients.

Tractor with Trailer	Drag [N]	Drag Coefficient
Original	553.05	0.54
O	370.75	0.32
A	361.01	0.31
B	347.94	0.30
C	332.70	0.28
D	318.05	0.27

**Table 6 biomimetics-08-00175-t006:** Model shapes vs. Drag coefficients.

Tractor	Drag [N]	Drag Coefficient
Original	651.50	0.634
O	423.65	0.359
A	411.58	0.350
B	386.35	0.327
C	390.60	0.328
D	400.98	0.341

**Table 7 biomimetics-08-00175-t007:** Model shapes with trailer.

Tractor and Trailer	Drag [N]	Drag Coefficient
Original	562.36	0.547
O	395.13	0.335
A	388.26	0.330
B	364.63	0.308
C	339.12	0.287
D	338.69	0.286

**Table 8 biomimetics-08-00175-t008:** Model shapes without trailer.

Tractor	Drag [N]	Drag Coefficient
Original	663.33	0.646
O	437.39	0.372
A	428.72	0.364
B	407.88	0.346
C	384.81	0.326
D	398.04	0.338

**Table 9 biomimetics-08-00175-t009:** Model shapes vs. total drag and skin friction drag.

Tractor and Trailer	Drag [N]	Friction [N]
Original	562.36	20.760
O	395.13	29.973
A	388.26	32.820
B	364.63	33.926
C	339.12	33.895
D	338.69	33.453

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
