# Peer review of "Bio-Model Selection, Processing and Results for Bio-Inspired Truck Streamlining"

_biomimetics, 2023, doi:10.3390/biomimetics8020175_

Round 1

Reviewer 1 Report (Previous Reviewer 3)

Thanks a lot for taking time for a careful revision. I would like to recommend its publication.

Author Response

Thank you for your efforts and for your approval.

Reviewer 2 Report (New Reviewer)

My concerns have been addressed.

Author Response

Thank you for your efforts and for your approval.

Reviewer 3 Report (New Reviewer)

The quality of the presentation needs serious improvement. The argument of the section needs improvement, and there is a lot of repetition of the information for example repeating the definition of the Reynolds number more than once which is a well-known number of the people working in this field. Please review my comments marked on the manuscript.

Author Response

Thank you for your efforts and for your comments and suggestions that improved the MS. More detailed responses to your comments can be found in the attached document.

This manuscript is a resubmission of an earlier submission. The following is a list of the peer review reports and author responses from that submission.

Round 1

Reviewer 1 Report

This is an interesting study on fish-inspired modifications of the front of trucks to reduce drag.  The model selection of demersal swimmers operating around the same Reynolds number was convincing to me, and a strong asset of the manuscript.  The CFD results are presented as ‘preliminary’ awaiting flow tunnel validations. As you will read in my numbered comments below, I have a few technical comments on the CFD.  This may require a few additional models to be calculated to ensure the results are valid.

(1): table 2: is this data for trucks? What is the load? This information seems needed to interpret these data correctly.

(2) line 115-122 & 2.2 species selection. I would like to read a little more framing on head function in fishes, as the head is not only used as a bow for swimming. Its shape can be a compromise between different functions (e.g. feeding, respiration, vision, etc.).

(3) formulae lines 164 & 170. For the middle parts with the values for each input variable, please add units.

(4) Reynolds number. Perhaps it can be mentioned that rainbow trouts can reach speeds as high as 2.72 m/s giving Re around 10^6.

(5) CFD – ground boundary condition. I understand that you are mimicking the flow tunnel of the experimental work in a flow tunnel that will follow in a later study. However, note that the ground moves backwards from the truck’s perspective, so the no-slip stationary boundary condition of your ‘flow splitter’ serving as ground is not realistic. Please acknowledge this effect, and write why you chose not to go for a more realistic BC (either flow allowed to slip, or wall moving with the air). And please, run one simulation with a slip BC, and evaluate the difference.

(6) CFD – flow domain size.  This size is small. CFD handbooks recommend at least 2 times the length of the object upstream and 5 times the length downstream. With a 8 m truck, a typical domain length would be minimally 64 m (instead of your values of 20 and 30 m; tables 5 and 6).  The problem is actually shown in Fig 15 where there is a still a strong wake pattern at the outflow boundary where you enforced a zero pressure change.  This is risky. Please run at least one simulation at larger domain size to ensure that this does not influence the results.

(7) tables 5 and 6: Why two simulations at the same Re would require a different flow domain size (table 5 and 6) is a mystery to me.

(8) CFD – simulation with water and air at the same Reynolds number should theoretically give exactly the same flow pattern, and hence the same drag coefficient.  Are the differences due to the different domain length and width? Any other reason you can think of?

(9) CFD mesh – conducting a mesh convergence analyses to ensure sufficient precision is a standard procedure for CFD. Please conduct and report at least one run with a slightly coarser mesh to ensure your mesh is fine enough.

(10) tables 7 to 10: please add to the legend whether this is in water or air.

(11) too few solver settings are reported to enable reproducing the reported results. Things like solver algorithms, incoming turbulence intensities, iterative convergence criteria should be listed.

(12) line 503 ‘users must have thorough fluid mechanics as well as numeric knowledge’. This sets the standards high for the current study. Given my above criticisms, I’m unsure whether you want to make such a claim.

(13) Discussion: how does the reported drag compare to more modern ‘aero’ designs? I heard about a Tesla truck with Cd = 0.36.  

Author Response

Thank you so much!

Reviewer 2 Report

The manuscript titled "Bio-Model Selection, Processing and Results for Bio-inspired Truck Streamlining" focuses on the CFD study aiming to decrease the aerodynamic resistance of the truck cabin by modifying its shape to resemble a trout's head. This topic is interesting. However, due to the flaws in how the utilized CFD model was described, I need to recommend rejecting the manuscript. I encourage the authors to implement the below suggestions and submit the manuscript again.

In order to be accepted, the manuscript needs to include crucial information regarding the performed CFD simulations. The manuscript lacks a mesh sensitivity study, which proves that the results are not affected by the mesh size. Another crucial element is validation. If the manuscript's authors do not provide experimental data to confirm that CFD simulations are correct, a comparison should be made with the appropriate experimental data from the literature. There is also no information about the mesh size, such as the total number of cells, boundary layer, and type of cells used to generate the final mesh. Without the data mentioned above, it is impossible to tell whether the data presented in the manuscript is correct. It also needs to be shown or described how far the model is from the inlet and outlet. A view of the complete computational domain and the view presenting the generated mesh are needed. There is no information on the turbulence on the inlet to the computational domain. For each studied geometry, more visualizations must be presented, for example, contours of pressure and graphs presenting pressure distribution along the truck cabin.

               Another remark refers to the methodology that was applied. Data for the truck with a trailer, presented in Figure 15, was obtained in air, whereas data for the truck without a trailer, presented in Figure 17, was obtained in water. It is incoherent to perform simulations in different conditions without explaining why such a choice was made. If this study uses only CFD simulations, it would be the most obvious to simulate the truck standing on the road and apply air as the fluid medium.

Figure 17 shows that the splitter plate on which the trailer is placed is too short, which influences the aerodynamic wake. Since these are numerical investigations, the splitter plate should be extended to the rear to eliminate this phenomenon.

Because the original shape of the trout was modified, it would also be interesting to present how these initial modifications influenced the trout's aerodynamic coefficients.

One more generic note is about the language used in the manuscript. For example, in the second paragraph, there is a sentence: "A reduction in vehicle drag will most probably contribute to the fuel saving and a decrease of air pollution, probably almost independent of vehicle purpose [6]." Instead of using such phrases as "most probably" or "probably," exact data should be provided.

Author Response

Thank you so much!

Reviewer 3 Report

This paper designed the shape of the truck’s tractor based on the trout’s head to reduce aerodynamic drag. The research method and object are not novel enough, some suggestions are as below:

1. As mentioned on Page 2, Line 69, the skin friction drag is hardly varying with the shape of the truck. The drag reduction method of this manuscript is to design a biological shape, which means the shape of the truck’s tractor has been changed. Although the friction drag barely changes during the increases of the truck’s running speed, the reviewer believes the change of the tractor’s shape changes the flow field around the whole truck body, it’d be better if the author represents the change of friction drag of the truck.

2. In Fig. 10, the “a” and “b” need to be marked beside the pics.

3. The details of the meshing section need to be represented. Pictures of the grids are necessary.

4. In section 3.2, the contour pictures of only one case are not enough, the pics of all cases are definitely necessary, which helps the reader to see the differences in the flow filed between all cases.

5. Since the comparison parameter is Cd, the pictures of the pressure distribution of the truck surface are necessary. In addition, there is maybe an extra line in the right pic of Fig. 15. If it is not, please explain. Finally, please note that the color of the slices is different from the legend.

6. The viewer can’t find the definition of Cd. The air resistance contains the pressure drag and the friction drag as indicated in the first part of the manuscript. However, the authors only mentioned “the drag and the drag coefficient” in section 3.2, the pressure drag was not discussed. If “the drag” stands for the pressure drag, the author needs to declare it at the beginning of section 3.2.

7. The creativity of this manuscript needs to be stated.

Author Response

Thank you so much!

Round 2

Reviewer 1 Report

Thank for addressing my comments. I am satisfied by this revision.

Author Response

Thank you very much!

Reviewer 2 Report

The quality of the manuscript has significantly improved, and I appreciate the work put in by the authors, but I believe further improvements are needed for the manuscript to be published. The description of the CFD model needs to be more precise, and the part of the manuscript presenting the CFD results should be rewritten.

The manuscript focuses on the results of CFD investigations, and to be published in any scientific journal, it needs to include all the essential information of the utilized computational model that I mentioned in the previous review. There is still no information in the manuscript on how many layers (if any) of the boundary mesh were generated in the computational grid. An adequate number of layers must be included for the boundary layer to be modeled correctly, and a specific value of the y+ parameter must be obtained. It is written that "The wall treatment condition was set as the wall function default," but it needs to be explained what the defaults are in the used CFD software. That is why it cannot be evaluated if the CFD model was prepared correctly. In the case of the results for the original model provided by the authors in the revised version of the manuscript, the results of drag force changed by 23%, which means that the authors already had to improve the results and with more focus on the boundary layer these results might change again.

The presented results of all modified models are almost the same, which raises the question of their point if they do not pose a further significant decrease of drag. The authors did not validate their CFD model or estimate the simulation error, which does not allow access to assess whether the change of drag coefficient from 0.30 to 0.28 is not within the calculation error and is meaningful. For example, the uncertainty of the numerical calculations described in "I. Celik, U. Ghia, Roache P. Christopher Procedure for estimation and reporting of uncertainty due to discretization in CFD applications J Fluids Eng, 130 (2008), Article 078001, 10.1115/1.2960953" could be utilized. If the authors focus on changes of this kind of magnitude that they need to prove that their model is accurate enough for this change to be meaningful.

The analysis of results needs to be more thorough than just focusing on the value of drag force. The contours of velocity need to be more thoroughly described and shown in more detail. Especially concerning the difference between the different cases, and if there are no differences, then it means that the proposed shape modifications have no meaningful consequences.

The Mesh study paragraph has no information about the number of elements in the fine mesh. Without it, it is impossible to say that mesh size does not affect the results.

All the above-described issues must be addressed for the manuscript to be considered for publication. Below are some of the other comments:

The top side of the splitter was set to "sliding wall." Will future experiments take place on a conveyor belt? If no, then a non-slip condition should be applied.

The simulations were done using a RANS turbulence model, and a symmetrical case was analyzed, which means that only half of the geometry could be used in the computations to decrease the size of the generated mesh and, thus, the computational time.

The authors wrote that all the presented calculations are now obtained for air. However, on Page 15, it is written otherwise: "The simulations were performed both with water and with air as the medium, because in a follow-up study, we will test physical models in a flow tank as well as in a wind tunnel [32]."

On Page 15, it is written, "Standard normal meshing was applied in our simulations." What is the meaning of the term "standard normal meshing"? This sentence should be dropped, and the authors should follow with the description of the mesh that they generated.

On Page 17, it is said that the results are presented in top view, but what is the location of the plane with velocity contours with regards to the truck or the splitter? Moreover, why this specific plane was chosen to present the results?

The bar graph presented on Page 19 is unnecessary. Apart from the original design, there are no differences in the results.

The rounded nose designs presented by the authors will generate lift force on the truck cabin. How is it going to affect the stability of the truck?

Author Response

Thank you very much!

Reviewer 3 Report

Where is your separate document of "point-by-point response to reviewer comments"? 

Author Response

I saw you found the letter, thanks.

Round 3

Reviewer 3 Report

This paper can be accepted now.

Author Response

Thank you very much!